# The Budapest Amyloid Predictor and Its Applications

**DOI:** 10.3390/biom11040500

**Published:** 2021-03-26

**Authors:** László Keresztes, Evelin Szögi, Bálint Varga, Viktor Farkas, András Perczel, Vince Grolmusz

**Affiliations:** 1PIT Bioinformatics Group, Eötvös University, H-1117 Budapest, Hungary; keresztes@pitgroup.org (L.K.); szogi@pitgroup.org (E.S.); balorkany@pitgroup.org (B.V.); 2MTA-ELTE Protein Modeling Research Group, H-1117 Budapest, Hungary; farkasv@caesar.elte.hu (V.F.); perczel@chem.elte.hu (A.P.); 3Laboratory of Structural Chemistry and Biology, Eötvös University, H-1117 Budapest, Hungary; 4Uratim Ltd., H-1118 Budapest, Hungary

**Keywords:** amyloid, support vector machines, site-specific amyloidogenecity, Budapest Amyloid Predictor

## Abstract

The amyloid state of proteins is widely studied with relevance to neurology, biochemistry, and biotechnology. In contrast with nearly amorphous aggregation, the amyloid state has a well-defined structure, consisting of parallel and antiparallel β-sheets in a periodically repeated formation. The understanding of the amyloid state is growing with the development of novel molecular imaging tools, like cryogenic electron microscopy. Sequence-based amyloid predictors were developed, mainly using artificial neural networks (ANNs) as the underlying computational technique. From a good neural-network-based predictor, it is a very difficult task to identify the attributes of the input amino acid sequence, which imply the decision of the network. Here, we present a linear Support Vector Machine (SVM)-based predictor for hexapeptides with correctness higher than 84%, i.e., it is at least as good as the best published ANN-based tools. Unlike artificial neural networks, the decisions of the linear SVMs are much easier to analyze and, from a good predictor, we can infer rich biochemical knowledge. In the Budapest Amyloid Predictor webserver the user needs to input a hexapeptide, and the server outputs a prediction for the input plus the 6 × 19 = 114 distance-1 neighbors of the input hexapeptide.

## 1. Introduction

The primary structure of the proteins is characterized by their amino acid sequence. While the primary structure basically determines the spatial folding of the proteins, and, consequently, all chemical and biological properties of the given protein, inferring those properties from the amino acid sequence is a very difficult task. Here, we consider the amyloid predictors—tools, which tell us if a given amino acid sequence has or does not have the propensity to become amyloid.

Amyloids are misfolded protein aggregates [1,2], which—in contrast with the unstructured aggregates—have a well-defined structure, comprising parallel β-sheets [3,4]. Amyloids are present in numerous organisms in biology: for example, in healthy human pituitary secretory granules [5]; in the immune system of certain insects [6], the silkmoth chorion and some fish choria [7]; in human amyloidoses and several neurodegenerative diseases [8].

Most recently, on the analogy of the naturally occurring antiherpes activity of β-amyloids, synthetic amyloid peptides were developed, acting as amyloidogenic aggregation cores in certain viral proteins with high specificity [9]. This way, new amyloid-based antiviral pharmaceuticals can be developed in the very near future: the specific aggregation cores turn the viral proteins into insoluble amyloids. Consequently, potential amyloidogenecity may have direct pharmaceutical relevance.

Sequence-based amyloid predictors would help the understanding and the exploitation of the amyloid state of the proteins: instead of the difficult, costly, and slow wet-laboratory tests, we can use the predictor on thousands or millions of inputs for enlightening the amyloidogenecity of proteins. A very recent review [10] covers the sequence-based amyloid-predictors, applying different strategies like AGGRESCAN [11], Zyggregator [12], netCSSP [13], and APPNN [14], among others.

In the last several years, the six-amino-acid-long peptides have become a model of studying amyloid formation [15,16,17]. The reason for this is twofold: first, numerous evidence shows the biological relevance of amyloid-forming hexapeptides [15,18,19,20,21]; second, one can form $20^{6}=$ 64 million hexapeptides from the 20 amino acids, which is a large—but not too large—and rich space of model molecules, whose structures are less complex and, therefore, easier to be dealt with as larger model spaces.

The APPNN predictor applies a machine-learning approach by training on 296 hexapeptides, selected from various sources, then predicts if a given hexapeptide is amyloidogenic or not. For longer sequences, it screens six-amino-acid-long sliding windows in longer polypeptide-chains to predict if they would form amyloid structures.

In this contribution, we construct and present a Support Vector Machine (SVM) predictor for hexapeptides, with better accuracy (84%) than most of the neural network-based tools (see [14] for a tabular comparison of the accuracy of those tools). We note that we do not repeat the comparative data described in [14], which evaluates numerous earlier published amyloidicity-prediction methods with APPNN. The main advantages of our new predictor, compared with other amyloid-predictors, are as follows:(i)Simplicity: we used solely a linear SVM in its construction;(ii)Transparency: no prefiltering and data manipulation were used in the construction of the predictor;(iii)Truly experimental training set: The experimental hexapeptide Waltz database [15,16] was applied in the SVM training and data were not privately filtered, predicted, and constructed as in other predictors;(iv)Free online availability, together with automatic prediction of the neighboring hexapeptides;(v)Easy applicability for inferring location-dependent amyloidogenic properties of amino acids, as we describe below.

We also note that neural-network-based predictors are neither simple nor easy to apply, and inferring the causality of their classifications is a very difficult task. In the case of SVMs, especially for linear SVMs, the causality is much more transparent, as we demonstrate in Table 1 and Table 2.

## 2. Methods

For the construction of the Budapest Amyloid Predictor, we have applied an artificial intelligence tool, the linear Support Vector Machine architecture [22], abbreviated as SVMs. In linear Support Vector Machines, n+m data points correspond to n+m vectors, each of *k* dimensions, $x_{1},x_{2},\;\ldots ,\;x_{n}$, and $y_{1},y_{2},\;\ldots ,\;y_{m}$, and the goal is to find a hyperplane that optimally separates the *x* and the *y* data points.

The mathematical foundation of SVMs is the trivial observation that any subset of the k+1 vertices of a *k*-dimensional simplex can be separated by a hyperplane from its complementer: it is obvious in a case of a triangle (a 2-dimensional simplex) or a tetrahedron (a 3-dimensional simplex). The mathematical problem becomes more interesting if the data points are not in general positions, or the separation is done in a smaller dimensional Euclidean space than the number of data points [23].

Usually, the dataset is partitioned into a training and a testing subset: the first one is applied in the construction of the SVM, the second one is used for testing the resulting tool.

We have used the Waltz database [15,16] of 1415 hexapeptides, annotated to be amyloidogenic (514 peptides) or nonamyloidogenic (901 peptides). The annotation in the Waltz database was made by Thioflavin-T binding assays and literature search [15,16]; consequently, it is based on experimental evidence. Similarly, as in [14], two vectorial representations of the hexapeptides were considered in the present work. The first is the simple translation of the 20 amino acid names into vectors, each amino acid X was corresponded to a length-20 0-1 vector, with a single 1-coordinate identifying X (called orthogonal representation). This way, a hexapeptide is described by a 6×20=120-dimensional 0-1 vector.

The second one is based on the AAindex, a physicochemical property database of 553 properties [24]; https://www.genome.jp/aaindex/ (accessed on 25 March 2021). In this representation, each amino acid corresponds to a 553-dimensional vector, and each hexapeptide to a 6×553=3318-dimensional vector.

From the 1415 (514 amyloid-, 901 nonamyloid-) hexapeptides found in Waltz database, we selected 158 amyloid and 309 nonamyloid hexapeptides randomly for the test set (roughly 33%). We used the remaining hexapeptides for training our linear SVM. We used the sklearn LinearSVC object from the SciKit-learn Python library [25] for constructing the classifier.

The orthogonal representation yielded approximately 80% accuracy, while the AAindex-based produced a much better accuracy; because of this, we have chosen the second, AAindex-based representation in what follows.

The classifier simply computes the sign of the w·z+b values for the 3318-long *z* vectors, corresponding to a hexapeptide, where *w* is a 3318-dimensional weight vector and *b* is a scalar; if this sign is positive, then the prediction is “amyloidogenic”, otherwise, it is “nonamyloidogenic”.

On the 467 test examples, we achieved 127 true positives, 31 false positives, 266 true negatives, and 43 false negatives. The resulting classifier’s performance for unseen examples is 0.8415 ± 0.0331 with 95% confidence. Based on test performance: ACC = 0.84, TPR = 0.75, TNR = 0.9, PPV = 0.8, NPV = 0.86, (that is, accuracy, true positive ratio, true negative ratio, positive predictive value, negative predictive value, respectively). The accuracy of our SVM is better than or on par with that of APPNN [14].

Figure 1 shows the ROC (Receiver Operating Characteristics) curve of the Budapest Amyloid Predictor. The AUC (Area Under Curve) value is 0.89. The precision-recall curve is provided as Appendix A.

The Budapest Amyloid Predictor uses the above mentioned SVM. For verifying the underlying method on differently selected random training and test-sets, we used 10-fold cross validations, with the construction of 10 distinct SVMs, detailed in the online Appendix A. The accuracies of the 10 SVM models were between 73% and 86% (see Appendix A). The analogs of our Table 1 for these SVMs are listed as Appendix A.

## 3. Discussion and Results

The Budapest Amyloid Predictor webserver is available at the site https://pitgroup.org/bap/ (accessed on 25 March 2021). The user needs to input a hexapeptide with 6 capital letters, and the server returns the prediction for the query, plus the predictions of all 114 (= 6 × 19) 1-Hamming-distance neighbors of the query. If the hexapeptide is listed in the Waltz DB, then the “known“ word appears next to prediction; otherwise, the “predicted” word appears.

Generally, it is very difficult to follow what a deep neural network does with a given input. In the case of a linear SVM, it is straightforward: for input vector *z*, the w·z+b quantity is computed, where the weight vector *w* and the scalar *b* are computed in the course of the SVM construction, and the sign of this quantity determines the prediction. Instead of specifying the 3318-dimensional weight vector *w* here, we present a very compact representation of the SVM in the next section. This representation not only specifies the predictor in a very simple way, but it also opens up novel insights of the amyloidogenic and nonamyloidogenic hexapeptides.

One of the greatest advantages of the linear SVM predictions is that we can easily see the reasons behind the decision of the model. If our model is accurate enough, then, from the coefficients of the normal vector of the separating hyperplane, the weight-differences of the distinct variables can be derived. We apply this observation below.

The following matrix enlightens the details of the decision of the linear SVM. Clearly, representing every amino acid by a 553-dimensional vector is highly redundant, since we have only 20 amino acids—that is, only 20 different 553-dimensional vectors exist in this representation. Therefore, we can write with ℓ=553:(1)$$w\cdot z=\sum_{i=1}^{6\ell }w_{i}z_{i}=\sum_{j=1}^{6}\;\;\sum_{i=(j-1)\ell +1}^{j\ell }w_{i}z_{i}.\;$$

For each fixed j=1,2,…,6, the ℓ=553$z_{i}^{\prime }$s are determined by the jth amino acid of the hexapeptide, this way, all the possible 6x20=120 second sums (for six positions and 20 amino acids) can be precomputed. Table 1 lists these precomputed values, the 6 values of *j* correspond to the columns and the amino acids to the rows.

The value of (1) can now be easily computed by adding exactly one item from each column, determined by the first, second, ..., sixth amino acid of the hexapeptide, plus the value of b=1.083. For example, one can classify the hexapeptide AAEEAA by computing the sign of (−0.26−0.32−0.43−0.30−0.43−0.22+1.083)=−0.88, that is, −1, which predicts that AAEEAA is not amyloidogenic.

By observing Table 1, one can easily derive an amyloidogenecity order of the amino acids for each position from 1 through 6: we just sort the columns in decreasing order and substitute the amino acids in the rows of Table 2 (from left to right). For example, in the first column of Table 1, the largest number corresponds to V, the second largest to I, so the first element of the first row of Table 2 is V, the second is I, and so on.

In Table 2, the amyloidogenecity order decreases from left to right.

The hydrophobic amino acids valine, isoleucine, phenylalanine, tyrosine, and tryptophan populate the left portion of Table 2, these amino acids are naturally more probable to form amyloids. Interestingly, alanine is not in that region, while cysteine is there.

Aspartic acid, lysine, asparagine, and glutamic acid populate the right end. Naturally, proline, the “structure breaker”, appears mostly at the right end, as one of the least amyloidogenic amino acids, but not in every row: in row 5, it is in position 12.

If there were no site-specific amyloidogenic properties of the amino acids, then all columns of Table 2 would be homogenic, i.e., every column would contain the same amino acid. Proline seems to be more “amyloid-breaker” in the ends and in the center of the hexapeptides, while much less so in the second and fifth position.

This table shows a remarkable difference in the amyloidogenecity order of the six positions of the hexapeptides: we believe that Table 2 is the most striking application of the Budapest Amyloid predictor.

### Comparison with Earlier Work

The location-dependent amyloidogenic properties of amino acids in hexapeptides were studied earlier in a conference paper [26] and the position-specific amyloidogenic properties of the amino-acids were listed in a Table (Table 1 in [26]). We note that our Table 2 is substantially different from that list: we order the 20 amino acids in amyloidogenic order in each of the 6 positions of hexapeptides. Additionally, in Table 2, proline has a very distinct structure-breaker property, while no such observation was found in Table 1 of [26].

The work [26] applied 139 amyloid and 168 nonamyloid peptides (after a nondetailed filtering procedure) for statistical analysis of amino acid frequencies in the positions of hexapeptides (see Figure 2 and Table 1 in [26]). We used a much larger dataset (514 amyloidogenic and 901 nonamyloidogenic hexapeptides) and our method is not simple frequency analysis, but a much deeper artificial intelligence approach.

Using SVMs for amyloid prediction is not without precedence: In [27], a nonlinear SVM is constructed for hexapeptide amyloid prediction. Instead of using experimentally identified amyloidogenic hexapeptides, the authors of [27] constructed an in-house “Hexpepset dataset”, where the positive and negative hexapeptides were gained from six amino acids sliding windows of known amyloid and nonamyloid proteins. We note that we use the experimentally verified Waltz dataset [15,16] of hexapeptides for training our linear SVM (each hexapeptide in the Waltz DB is annotated experimentally and not by theoretical inference with sliding windows of known amyloids or nonamyloids). Additionally, we attain a 84% accuracy on the experimental Waltz dataset, compared to the 81% accuracy of the much more complex, nonlinear SVM of [27] on a theoretically (sliding windows) and ad hoc constructed, nonexperimental, “Hexpepset dataset” of hexapeptides.

## Figures and Tables

**Figure 1 biomolecules-11-00500-f001:**
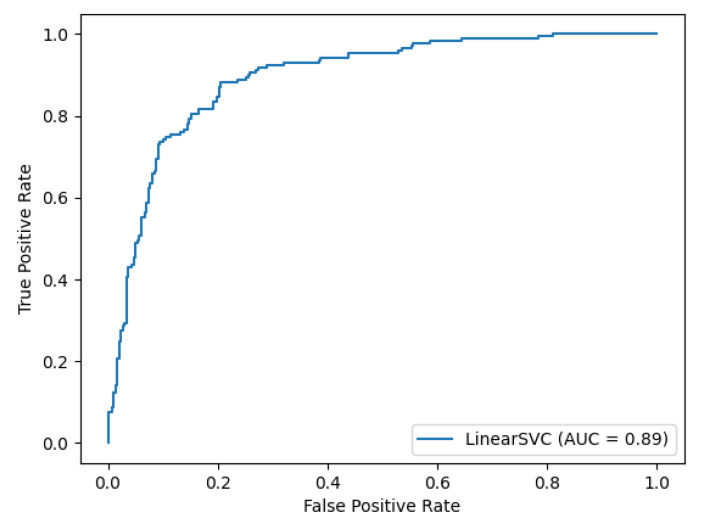
The ROC (Receiver Operating Characteristics) curve of the Budapest Amyloid Predictor. The AUC (Area Under Curve) value is 0.89. The precision-recall curve is provided as Appendix A.

**Table 1 biomolecules-11-00500-t001:** The precomputed values from Equation (Equation 1) are listed in the rows, corresponding to the amino acids. The columns are correspond to their positions.

	1	2	3	4	5	6
A	−0.26	−0.32	−0.27	−0.14	−0.43	−0.22
R	−0.45	−0.41	−0.46	−0.33	−0.52	−0.35
N	−0.40	−0.34	−0.49	−0.27	−0.46	−0.30
D	−0.49	−0.43	−0.56	−0.41	−0.56	−0.36
C	−0.09	−0.21	0.03	−0.05	−0.17	−0.05
Q	−0.37	−0.30	−0.36	−0.34	−0.48	−0.32
E	−0.51	−0.41	−0.43	−0.30	−0.61	−0.39
G	−0.23	−0.37	−0.46	−0.37	−0.30	−0.33
H	−0.32	−0.26	−0.26	−0.30	−0.35	−0.25
I	−0.06	−0.08	0.26	0.09	−0.06	−0.07
L	−0.10	−0.18	0.02	0.04	−0.22	−0.13
K	−0.39	−0.45	−0.51	−0.35	−0.59	−0.32
M	−0.17	−0.25	−0.02	−0.10	−0.19	−0.18
F	−0.13	−0.11	0.05	−0.03	−0.13	−0.11
P	−0.56	−0.38	−0.56	−0.51	−0.42	−0.45
S	−0.37	−0.35	−0.41	−0.30	−0.48	−0.23
T	−0.34	−0.33	−0.28	−0.23	−0.40	−0.23
W	−0.17	−0.17	−0.09	−0.06	−0.12	−0.16
Y	−0.23	−0.11	−0.13	−0.06	−0.18	−0.15
V	−0.05	−0.14	0.19	0.14	−0.19	0.01

**Table 2 biomolecules-11-00500-t002:** The amyloidogenecity order of the amino acids, decreasing from left to right.

	1	2	3	4	5	6	7	8	9	10	11	12	13	14	15	16	17	18	19	20
1	V	I	C	L	F	M	W	G	Y	A	H	T	S	Q	K	N	R	D	E	P
2	I	F	Y	V	W	L	C	M	H	Q	A	T	N	S	G	P	R	E	D	K
3	I	V	F	C	L	M	W	Y	H	A	T	Q	S	E	R	G	N	K	D	P
4	V	I	L	F	C	W	Y	M	A	T	N	H	E	S	R	Q	K	G	D	P
5	I	W	F	C	Y	M	V	L	G	H	T	P	A	N	Q	S	R	D	K	E
6	V	C	I	F	L	Y	W	M	A	T	S	H	N	Q	K	G	R	D	E	P

## Data Availability

The Budapest Amyloid Predictor webserver is available freely at https://pitgroup.org/bap (accessed on 25 March 2021).

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
