# Peer review of "The Budapest Amyloid Predictor and Its Applications"

_biomolecules, 2021, doi:10.3390/biom11040500_

Round 1
Reviewer 1 Report
The authors present an SVM based method to distinguish amyloidogenic hexapeptide sequences from non-amyloidogenic ones.
While the problem is important and the method has merit, the work is still very preliminary. It is very specific to hexapeptides and was trained on a relatively small dataset. Can it be scaled to other sizes?
What does the number b mean and how was its value determined?
How can the values in table 1 be interpreted? Are these log probabilities?
How do the results explain the fact that in many cases (such as the Huntington's protein), poly-Q repeats are responsible for amyloid generation?
Author Response
See the enclosed file.

Reviewer 2 Report
The authors introduced a linear support vector machine (SVM)-based predictor for hexapeptides. The followings are my comments to be addressed by the authors:
1. The authors need to perform 5-fold or 10-fold cross-validation in order to evaluate the performance of the proposed model
2. The authors need to discuss or investigate other properties like PSSM, SS properties, etc., and other machine learning techniques.
3. In the case of imbalanced data, the authors are recommended to provide the area under the precision-recall curve or MCC
4. A case study should be provided on recently added data in the literature (if any) in order to test the generalization of the proposed model.
Author Response
See the enclosed file.

Reviewer 3 Report
In the manuscript, Keresztes et al., introduces a new Support Vector Machine (SVM)-based predictor for amyloidal hexapeptides. This new model improves on its predecessors in terms of accuracy, simplicity, usability, and experiment-based database learning. The paper is well-written and explains the development of the predictor clearly, and its findings will be significant contribution to the field of computational protein prediction.
Please address the following concerns prior to publication:
- Compared to other amyloid predictors, Keresztes’s is more user-friendly and intuitive to use with clear instructions provided at the top of the website. In terms of graphic design, the website is clean and uncluttered. A suggestion to improve the website’s usability would be to incorporate color to distinguish between amyloidal or non-amyloidal results.
- The authors wrote in the introduction, “…the primary structure determines the spatial folding of proteins.” This statement is an oversimplification as the environment of a protein is extremely relevant to its structure and folding1. The authors should address or revise this claim prior to publication.
- This software’s machine learning was based on the Waltz database, but upon testing with other hexapeptides from other databases (FoldAmylod and AmylPred), the outputs have shown that this predictor had low accuracy. Similarly, when testing this software with experimentally proven amyloidogenic cores of nine peptides or longer, even with sliding frames, the software was not accurate in predicting amyloidogenicity.
- The goal of this paper was to introduce a new predictor based on a simpler learning algorithm that can more accurately predict the amyloidal properties of hexapeptides. While the authors ultimately succeeded in doing so, the outputs indicate that the predictor has room to improve, perhaps with further training from a larger dataset pool.
Citation:
- ANFINSEN CB, HABER E. Studies on the reduction and re-formation of protein disulfide bonds. J Biol Chem. 1961 May;236:1361-3. PMID: 13683523.
Author Response
See the enclosed file.

Round 2
Reviewer 1 Report
The authors responded to the reviewers' comments. It is still somewhat unclear how the values in Table 1 were computed. The formula is clear (eq. 1) but what were the initial values, how were the initial values for b and w were determined, plus a reference to what I believe is the Sup. material tables.
As it is now, it is not possible to reproduce the results for lack of sufficient details.
Author Response
Please find enclosed.

Reviewer 2 Report
The reported performance should be based on 10-fold cross-validation. For example, the accuracy of the proposed model is not 0,84. Instead, it is 0,8084+/- 0.0412
Please add captions to the tables.
In the previous review and as the data imbalanced, I asked the authors to show the area under the precision-recall curve in addition area under the receiver operating characteristic curve. However; the authors did not provide it.
Author Response
Please find enclosed.
